# Potential Renewable Hydrogen from Curtailed Electricity to Decarbonize ASEAN's Emissions: Policy Implications

**Han Phoumin [1,\*], Fukunari Kimura [1,2] and Jun Arima [1,3]**

[1] Economic Research Institute for ASEAN and East Asia (ERIA), Think Tank, Jakarta 10270, Indonesia; vzf02302@nifty.ne.jp (F.K.); junarima@g.ecc.u-tokyo.ac.jp (J.A.)
[2] Faculty of Economics, Keio University, Tokyo 108-8345, Japan
[3] Graduate School of Public Policy, Tokyo University, Tokyo 113-0033, Japan
[\*] Correspondence: han.phoumin@eria.org

**Abstract:** The power generation mix of the Association of Southeast Asian Nations (ASEAN) is dominated by fossil fuels, which accounted for almost 80% in 2017 and are expected to account for 82% in 2050 if the region does not transition to cleaner energy systems. Solar and wind power are the most abundant energy resources but contribute negligibly to the power mix. Investors in solar or wind farms face high risks from electricity curtailment if surplus electricity is not used. Employing the policy scenario analysis of the energy outlook modelling results, this paper examines the potential scalability of renewable hydrogen production from curtailed electricity in scenarios of high share of variable renewable energy in the power generation mix. The study found that ASEAN has high potential in developing renewable hydrogen production from curtailed electricity. The study further found that the falling cost of renewable hydrogen production could be a game changer to upscaling the large-scale hydrogen production in ASEAN through policy support. The results implied a future role of renewable hydrogen in energy transition to decarbonize ASEAN's emissions.

**Keywords:** energy transition; renewables; hydrogen; fossil fuels; emissions

## 1. Introduction

The economic, social, and political dynamics of the Association of Southeast Asian Nations (ASEAN) have made it one of the fastest-growing regions. However, Southeast Asia faces great challenges in matching its energy demand with sustainable energy supply as the region transitions to a lower-carbon economy. The transition requires development and deployment of green energy sources. Growing energy demand can be met by energy supply produced by renewables and other clean energy alternatives such as hydrogen and by clean technologies [1]. Whilst Organization for Economic Co-operation and Development (OECD) countries have quickly reduced greenhouse gas emissions in response to the commitments of the Paris Climate Conference or the 21st Conference of the Parties (COP 21), developing Asia has some way to go to balance economic growth and affordable and available energy. Much of the future energy mix of emerging ASEAN countries will rely on fossil fuel to power economic development. However, they can follow a renewable energy path to economic growth, social well-being, and environmental sustainability.

The power generation mix of the Association of Southeast Asian Nations (ASEAN) is dominated by fossil fuels, which accounted for almost 80% in 2017 and are expected to account for 82% in 2050 if the region does not transition to cleaner energy systems [2].

Reducing greenhouse gas emissions is high on the global agenda under COP 21 and the upcoming COP 26, which will require leaders to pursue alternative fuel pathways, shifting from fossil fuel–based

to clean energy systems. In this regard, hydrogen fuel represents growth potential as world leaders start to see the great benefit and promise of its use to abate climate change. In many ASEAN countries, hydrogen as an alternative fuel is not yet on the policy agenda. The ASEAN Plan of Action for Energy Cooperation (APAEC) Phase 2, however, will include policy measures to encourage emerging and alternative technologies such as hydrogen and energy storage.

The potential use of hydrogen in transport, power generation, and industry has been proven by projects around the world. Renewable hydrogen has attracted leaders' attention as an option to increase the share of renewables in electrical grids amidst the falling cost of renewable electricity from wind and solar energy. The International Renewable Energy Agency (IRENA) [3] predicted that the cost of electrolyzers, the devices used to produce hydrogen from water, will halve from US$840 now to US$420 per kW by 2040. Renewable hydrogen production could be the cheapest energy option in the foreseeable future. The cost-competitiveness of producing renewable $H_2$ is key for the wide adoption of hydrogen. Renewable $H_2$ production costs dropped drastically from US$10–US$15/kg in 2010 to US$4–US$6/kg in 2020 [4]. Costs are expected to decrease to US$2.00–US$2.50/kg of $H_2$ in 2030, which is competitive with hydrogen production using natural gas through steam methane reforming with carbon capture, sequestration, and storage (CCS) [5].

Hydrogen is a clean energy carrier and can be stored and transported for use in hydrogen-run vehicles, synthetic fuels, upgrading of oil and/or biomass, ammonia and/or fertilizer production, metal refining, heating, and other end uses. Developing hydrogen, therefore, is an ideal pathway to sustainable clean energy systems and can help scale renewables such as solar and wind energy. Adopting renewable hydrogen would bring more renewables into the energy mix and could be a game changer in the transition from fossil dependence to a cleaner energy system in ASEAN. Hydrogen could help integrate the current electricity system with wind and solar energy. Solar and wind penetration of the electrical grid is hindered by the high intermittency of electricity from wind and solar energy, and many grid operators in ASEAN are, therefore, hesitant to include a large share of it.

The Economic Research Institute for ASEAN and East Asia's research on hydrogen energy since 2017 has identified the significant potential of hydrogen energy supply and demand in East Asia. By 2040, the cost of hydrogen will decrease by more than 50% if it is adopted in all sectors. The target price of US$2.00–US$2.50/kg of $H_2$ in 2040 is competitive with the price of gasoline. The cost of supplying hydrogen is about 3–5 times higher than that of gas, mainly due to limited investment in hydrogen supply chains and the lack of a strategy to widely adopt hydrogen usage. The wide adoption and usage of hydrogen will need time to ensure cost-competitiveness and safety, especially for automobiles. The large-scale hydrogen-based energy transition from 'grey' and 'blue' to 'green' hydrogen will happen concurrently with a global shift to renewables. 'Green' hydrogen can face current system integration challenges that have blocked increasing the share of wind and solar energy.

In ASEAN, Brunei Darussalam leads in the hydrogen supply chain and has supplied liquefied hydrogen from Muara port to Japan since late 2019 [6]. However, the liquefied hydrogen process consumes a great deal of energy to cool gaseous hydrogen into liquid hydrogen at temperatures of −253 degrees Celsius and lower. The hydrogen supply chain demonstration project, in cooperation with Japan's government, explored an alternative way of shipping hydrogen using a new technology called liquid organic hydrogen carrier. If the technology is economically viable, it will pave the way for market access worldwide and overcome hydrogen supply chain barriers.

In many ASEAN countries, hydrogen is not yet on the policy agenda as an alternative fuel. However, APAEC, which is under preparation for endorsement at the ASEAN Ministers on Energy Meeting in November 2020, will include policy measures to promote emerging and alternative technologies such as hydrogen and energy storage [7]. APAEC will help AMS increase their adoption of hydrogen to enlarge the share of hydrogen in the energy mix.

The study investigates the potential of renewable hydrogen as a clean energy source for ASEAN's energy mix, which will need huge investment in hydrogen energy–related industries. The paper aims to do the following:

1.  Use energy modelling scenarios to explore policy options of increasing the share of renewables, particularly wind and solar energy, in the power mix, and explore the possibility of electricity curtailment resulting from the high share of renewables that can be converted to hydrogen production.
2.  Estimate the potential emission abatement resulting from the introduction of hydrogen produced using curtailed renewable electricity.
3.  Review scalable renewable electricity from wind and solar energy from a cost reduction perspective, considering global experience.
4.  Review technologies and cost perspectives of hydrogen produced using curtailed electricity.
5.  Review a hydrogen policy and road map that can be applied to ASEAN.

Hydrogen adoption and development could be highly beneficial for ASEAN. Renewable hydrogen will enable the deployment of variable renewable energy (VRE) such as wind and solar and will be a game changer by breaking the barrier of integrated traditional power systems, which cannot absorb a high share of wind and solar energy. The paper is organized as follows: Section 2 reviews the pathways of hydrogen production processes; Section 3 explains the methodological approaches; Section 4 discusses the study's results; and Section 5 draws conclusions and policy implications.

## 2. Selected Pathways of Hydrogen Production Processes

Hydrogen emits zero emissions when used in combustion for heat and energy. If pure hydrogen ($H_2$) combusts by reacting with oxygen ($O_2$), it will form water ($H_2O$) and release energy that can be used as heat, in thermodynamics, and for thermal efficiency. Hydrogen is the most abundant chemical substance in the universe, but it is rarely found in pure form ($H_2$) because it is lighter than air and rises into the atmosphere. Hydrogen is found as part of compounds such as water and biomass and in fossil fuels such as coal, gas, and oil [8]. Several ongoing researches use two processes to extract hydrogen fuel: steam methane reforming, mainly applied to extract hydrogen from fossil fuels, and electrolysis of water, applied to extract hydrogen from water using electricity.

Steam methane reforming extracts hydrogen from methane using high-temperature steam (700–1000 °C). The product of steam methane reforming is hydrogen, carbon monoxide, and a small amount of carbon dioxide [9]. Most hydrogen is produced through this process, which is the most mature technology. Given how cheap natural gas is in the US and other parts of the world, hydrogen is one pathway to transition to a cleaner economy if steam methane reforming can be augmented with CCS. Technically, the chemical reaction process can be written as follows.

Steam methane reforming reaction (heat must be supplied through an endothermic process):

$$CH_4 + H_2O \ (+heat) \rightarrow CO + 3H_2, \tag{1}$$

Applying water-gas shift reaction (1) produces more hydrogen:

$$CO + H_2O \rightarrow CO_2 + H_2 (+small \ amount \ of \ heat), \tag{2}$$

At this stage, carbon dioxide and other impurities are removed from the gas stream, so the final product is pure hydrogen.

Instead of steam methane reforming, partial oxidation can be applied to methane gas to produce hydrogen. However, the partial oxidation reaction produces less hydrogen fuel than does steam methane reforming. Technically, partial oxidation is an exothermic process, producing carbon monoxide and hydrogen and giving off heat:

$$CH_4 + \frac{1}{2}O_2 \rightarrow CO + 2H_2 \ (+heat), \tag{3}$$

Applying a water-gas shift reaction in (3) produces more hydrogen:

$$CO + H_2O \rightarrow CO_2 + H_2 (+small\ amount\ of\ heat),\qquad(4)$$

Electrolysis can produce hydrogen by splitting water into hydrogen and oxygen in an electrolyzer, which consists of an anode and a cathode. Electrolyzers may have slightly different functions depending on the electrolyte material used for electrolysis.

The polymer electrolyte membrane (PEM) electrolyzer is an electrochemical device to convert electricity and water into hydrogen and oxygen. The PEM electrolyte is solid plastic. The half reaction that takes place on the anode side forms oxygen, protons, and electrons:

$$2H_2O \rightarrow O_2 + 4H^+ + 4e^-,\qquad(5)$$

The electrons flow through the external circuit and the hydrogen ions move across the PEM to the cathode, in which hydrogen ions combine with electrons from the external circuit to form hydrogen gases:

$$4H^+ + 4e^- \rightarrow 2H_2,\qquad(6)$$

PEM electrical efficiency is about 80% in terms of hydrogen produced per unit of electricity used to drive the reaction. PEM efficiency is expected to reach 86% before 2030.

Another method is alkaline water electrolysis, which takes place in an alkaline electrolyzer with alkaline water (pH > 7) with an electrolyte solution of potassium hydroxide (KOH) or sodium hydroxide (NaOH). In the alkaline electrolyzer, the two electrodes are separated. Hydroxide ions (OH−) are transported through the electrolyte from cathode to anode, with hydrogen generated on the cathode side. This method has been commercially available for many years, and the new method of using solid alkaline exchange membrane is promising as it is working in a laboratory environment.

## 3. Methodology and Scenario Assumptions

Hydrogen is used mainly to produce petrochemicals and ammonia. The potential of hydrogen, however, clearly remains untapped in ASEAN countries because it is a clean energy carrier that can be produced from various sources using fossil fuel and renewable energy. To build a hydrogen society, the cost of producing hydrogen must be competitive with that of conventional fuels, such as gas, for transport and power generation.

Renewable or 'green' hydrogen must be produced using renewable electricity from wind, solar, hydropower, and geothermal energy. Excess electricity from nuclear power, however, could be used to produce hydrogen as nuclear power plants provide base-load power and cannot be easily ramped up and down. During low demand, electricity from nuclear energy and VRE could be used to produce hydrogen. To produce renewable hydrogen using VRE, it is important to know the predicted available curtailed electricity resulting from power system integration challenges due to higher share of renewables.

Two components determine the cost to produce 'green' hydrogen: electricity cost from renewables and the cost of electrolysis. If these costs could be reduced significantly to allow the cost of hydrogen production to be competitive with that of natural gas, then hydrogen adoption and usage could be accelerated. This study reviews the falling cost of VRE and electrolysis to see how their current and future cost could allow a competitive hydrogen production cost. High VRE penetration of the electrical grid is the biggest challenge for the grid operator as electricity from VRE is variable and intermittent. Upgrading the grid system with the Internet of Things to create a smart grid could allow more penetration by VRE; otherwise, VRE electricity would be greatly curtailed due to a weak power grid system. This study calculates potential renewable hydrogen production and potential emission abatement under various scenarios assuming the following:

- Under current grid system integration, curtailment is likely to be 20–30% if the VRE share in the power mix exceeds more than 10%. Given the large potential of hydropower, geothermal, wind, and solar energy, increasing the share of renewables is technically possible using hydrogen storage. The study assumes the following scenarios: replacement by renewables of total combined fossil fuel generation (coal, oil, and gas) by 10%, 20%, and 30% by 2050, or, Scenario1 = 10%, Scenario2 = 20%, and Scenario3 = 30%.
- In Scenario1, Scenario2, and Scenario3, renewable hydrogen production using curtailed electricity is calculated based on assumptions of curtailed electricity generated from renewables at the rate of 20–30% of total generation from renewables. Potential renewable hydrogen produced using curtailed electricity in Scenario1, Scenario2, and Scenario3 is expressed as Scenario1$H_2$, Scenario2$H_2$, and Scenario3$H_2$.
- The formulas to calculate potential renewable hydrogen production in the renewable scenarios are as follows:

  ○　　Scenario1$H_2$ (Mt-$H_2$) = [Scenario1 (TWh) × (Percentage of curtailed electricity)/48 (TWh)].
  ○　　Scenario2$H_2$ (Mt-$H_2$) = [Scenario2 (TWh) × (Percentage of curtailed electricity)/48 (TWh)].
  ○　　Scenario3$H_2$ (Mt-$H_2$) = [Scenario3 (TWh) × (Percentage of curtailed electricity)/48 (TWh)].

Mt-$H_2$ stands for million tonnes of hydrogen; TWh is terawatt-hour; and percentage of curtailed electricity is 20–30% of total generation from renewables. The study also applies the conversion factor of 48 kilowatt-hours (kWh) of electricity needed to produce 1 kg $H_2$ [10].

The potential emission abatement is the difference between (a) the business as usual (BAU) scenario and (b) the alternative policy scenario (APS) and other high-renewable-share scenarios such as Senario1, Scenario2, and Scenario3.

To estimate potential hydrogen produced using curtailed electricity, the power generation mix for the BAU and APS is estimated using ASEAN countries' energy models by applying the Long-range Energy Alternative Planning System (LEAP) software, an accounting system to project energy balance tables based on final energy consumption and energy input and/or output in the transformation sector. The LEAP software has been chosen in this study to estimate the future demand in power generation mix because the input of energy data provided by experts from ASEAN member states adopted their energy demand and supply modelling based on the LEAP modelling structure. Thus, the forecast of power generation demand is based on energy demand equations by energy and sector and future macroeconomic assumptions.

In the modelling work applying LEAP, the baseline of 10 AMS was 2017, the real energy data available in 2017, which are the latest that the study employed. Projected demand growth is based on government policies, population, economic growth, and other key variables, such as energy prices used by the International Energy Agency energy demand model [11]. BAU is in line with current energy policy in the baseline information, which is used to predict future energy demand growth. However, APS differs from BAU in policy changes and targets, with a greater share of renewables, including possible nuclear uptake based on an alternative policy for energy sources and more efficient power generation and energy in final energy consumption.

For electricity generation, experts from 10 AMS specified assumptions based on their national power development plans and used the assumptions to predict ASEAN's power generation mix. For renewable hydrogen production, the study applies a conversion factor of 48 kWh of electricity needed to produce 1 kg of hydrogen [10].

## 4. Results and Discussion

The potential of renewable hydrogen produced using curtailed electricity in Scenario1, Scenario2, and Scenario3 is quantified according to a renewable curtailment rate of 20–30% for the high share of renewables in 2050. Emission abatement—the difference between (i) BAU and (ii) APS, Scenario1,

Scenario2, and Scenario3—is calculated. The higher share of renewables under Scenario1, Scenario2, and Scenario3 could only happen if hydrogen is developed as an energy storage by utilizing curtailed renewable electricity. The study discusses hydrogen as an enabler of higher shares of renewables, the need to reduce the cost of renewable hydrogen production by reducing the cost of electrolysis and renewables, and the need to develop a hydrogen road map for ASEAN to guide industry and key investors in renewable hydrogen development. The road map will help create a large-scale ASEAN hydrogen society.

### 4.1. Potential Renewable Hydrogen from Curtailed Electricity

ASEAN's power generation is dominated by fossil fuel (coal, oil, and gas), the share of which in the power mix was 79% (equivalent to 1041 TWh) in 2017 and is predicted to be 82% (2826 TWh) and 72% (2087 TWh) in 2050 for BAU and APS, respectively (Figure 1). The share of combined fossil fuel (coal, oil, and gas) in the power generation mix is expected to reduce drastically from 82% in BAU to 65%, 58%, and 51% in Scenario1, Scenario2, and Scenario3, respectively, in 2050 (Figure 2). The share of combined renewables is expected to increase from 18% in BAU to 35%, 42%, and 49% in Scenario1, Scenario2, and Scenario3, respectively, in 2050. The higher share of renewables in the power generation mix is desirable to decarbonize emissions in ASEAN's future energy system. However, the high share of renewables can only happen with bold policy actions to develop and deploy renewable hydrogen to support the power integration system, which has a higher penetration of renewables. Utilizing unused electricity and/or curtailed renewable electricity to produce hydrogen could be ideal to tap the maximum potential of renewables.

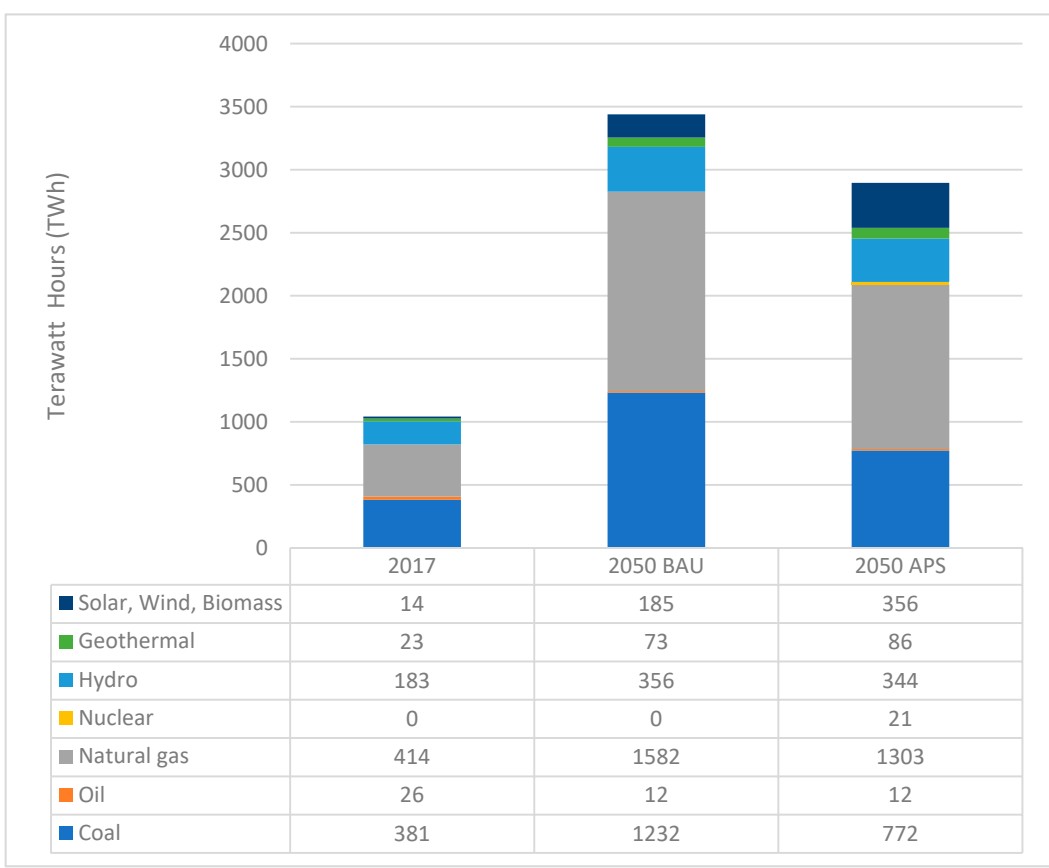

| | 2017 | 2050 BAU | 2050 APS |
|---|---|---|---|
| ■ Solar, Wind, Biomass | 14 | 185 | 356 |
| ■ Geothermal | 23 | 73 | 86 |
| ■ Hydro | 183 | 356 | 344 |
| ■ Nuclear | 0 | 0 | 21 |
| ■ Natural gas | 414 | 1582 | 1303 |
| ■ Oil | 26 | 12 | 12 |
| ■ Coal | 381 | 1232 | 772 |

**Figure 1.** ASEAN's Power Generation Mix in Business as Usual and Alternative Policy Scenario by Source. BAU = business as usual, APS = alternative policy scenario. Source: Authors.

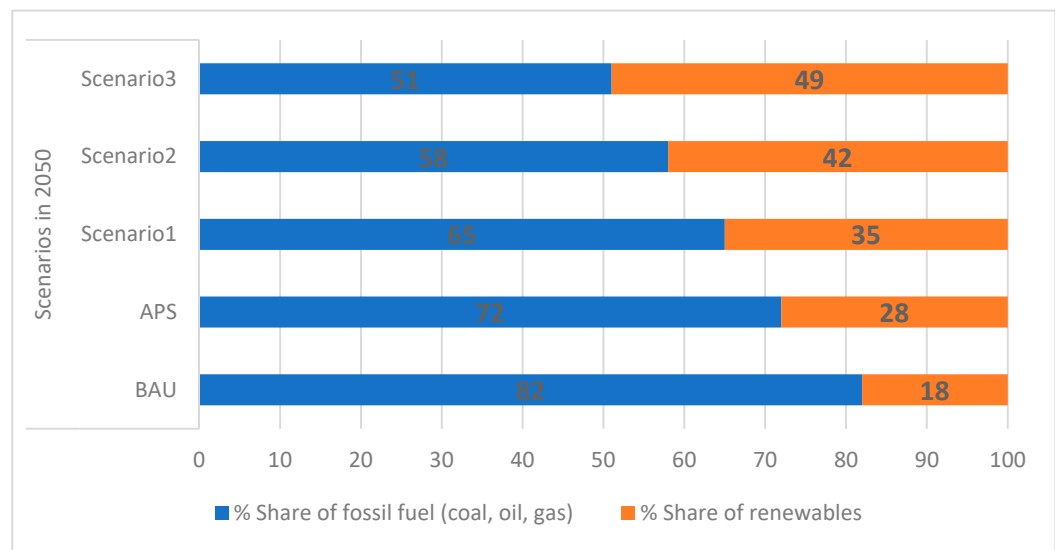

**Figure 2.** Share of Combined Fossil Fuels (coal, oil and gas) vs Renewables under Various Scenarios. APS= alternative policy scenario, BAU = business as usual. Note: Scenario1, Scenario2, and Scenario3 envision replacing combined fossil fuel (coal, oil, and natural gas) power generation with renewables (mainly variable renewable energy) at 10%, 20%, and 30%, respectively, in 2050. Source: Authors.

Scenario1, Scenario2, and Scenario3 assume the replacement of combined fossil fuel (coal, oil, and gas) power generation in 2050 with 10%, 20%, and 30% of power generation from renewables. Renewable power generation amounts in 2050 are 1016 TWh, 1224 TWh, and 1433 TWh for Scenario1, Scenario2, and Scenario3, respectively (Table 1).

**Table 1.** ASEAN's Power Generation Mix under Various Scenarios of Share of Renewables (TWh).

|  | 2050 APS | Replacement of Coal, Oil, and Natural Gas by Renewables | | |
|---|---|---|---|---|
|  |  | Scenario1 = 10% | Scenario2 = 20% | Scenario3 = 30% |
| Coal | 772 | 698.8 | 618 | 540 |
| Oil | 12 | 11 | 10 | 8 |
| Natural gas | 1303 | 1173 | 1042 | 912 |
| Renewables (wind, solar, hydro, geothermal, and/or biomass) | 807 | 1016 | 1224 | 1433 |

APS = alternative policy scenario. Note: Scenario1, Scenario2, and Scenario3 envision replacing combined fossil fuel (coal, oil, and natural gas) power generation with renewables (mainly variable renewable energy) at 10%, 20%, and 30%, respectively, in 2050. Source: Authors.

In Scenario1, Scenario2, and Scenario3, the shares of renewables in the power mix will be 35%, 42%, and 49%, respectively, in 2050. Because of higher shares of renewables in the power mix, renewable energy generation will be highly curtailed. The curtailed electricity rate could vary from 20% to 30%, depending on the power grid infrastructure in AMS. Based on this curtailed electricity, with varying shares of renewables in Scenario1, Scenario2, and Scneario3, hydrogen production scenarios are created— Scenario1$H_2$, Scenario2$H_2$, and Scenario3$H_2$. Potential renewable hydrogen from curtailed electricity in scenarios in AMS range from 4.23 to 8.96 million tonnes hydrogen (Table 2).

**Table 2.** ASEAN's Potential Renewable Hydrogen ($H_2$) from Curtailed Electricity.

| Hydrogen Production | Potential Renewable Hydrogen Production | | |
|---|---|---|---|
| | Scenario1$H_2$ (Million Tonnes $H_2$) | Scenario2$H_2$ (Million Tonnes $H_2$) | Scenario3$H_2$ (Million Tonnes $H_2$) |
| Of 20% curtailed renewables | 4.23 | 5.10 | 5.97 |
| Of 30% curtailed renewables | 6.35 | 7.65 | 8.96 |

$H_2$ = hydrogen, Scenario1$H_2$ = hydrogen production in Scenario1, Scenario2$H_2$ = hydrogen production in Scenario2, Scenario3$H_2$ = hydrogen production in Scenario3. Note: 20–30% curtailed electricity applied for combined renewable power generation in 2050. The study applied a conversion factor of 48 kilowatt-hours (kWh) of electricity needed to produce 1 kilogram (kg) $H_2$ [10]; 1 kg of $H_2$ could generate 33.3 kWh [10]. Source: Authors.

The higher share of renewables under various scenarios such as APS, Scenario1, Scenario2, and Scenario3 will see a large reduction in carbon dioxide emissions ($CO_2$), which could result in decarbonizing emissions and contribute to COP commitments. Potential emission abatement ranges from −340 million tonnes carbon (Mt-C) in APS to −648 Mt-C, −710 Mt-C, and −774 Mt-C in Scenario1, Scenario2, and Scenario3, respectively (Table 3). Emissions were cut by 28% from BAU to APS, 53% from BAU to Scenario1, 58% from BAU to Scenario2, and 64% from BAU to Scenario3.

**Table 3.** Potential Emission Reduction under Various Scenarios (Mt-C).

| | 2017 | 2050 | 2050 | 2050 |
|---|---|---|---|---|
| | Baseline | Emissions under Various Scenarios | Emission Abatement Potential | % Emission Reduction from BAU |
| BAU | 376 | 1216 | | |
| APS | 376 | 876 | −340 | 28% |
| Scenario1 | 376 | 568 | −648 | 53% |
| Scenario2 | 376 | 506 | −710 | 58% |
| Scenario3 | 376 | 442 | −774 | 64% |

APS = alternative policy scenario, BAU = business as usual, Mt-C = million tonnes carbon. Note: Emission abatement potential is change of emissions from BAU to APS and other scenarios in 2050 under high renewables in Scenario1, Scenario2, and Scenario3. Source: Authors.

### 4.2. Hydrogen, an Enabler to Scale up Variable Renewable Energy

In ASEAN, power generation is dominated by coal, gas, and hydropower. Intermittent renewables from solar and wind energy contributed a negligible amount (14.47 TWh) or about 1.4% in 2017. However, the most optimistic prediction is that ASEAN will increase the share of wind and solar energy in the power generation mix to about 12.3% by 2050 (calculated from Figure 1). The inclusion of the share of hydro (17.6%) and geothermal (2.2%) energy in the power generation mix contributed to the overall renewable share of 21.2% in 2017. However, future abundant resources are wind and solar energy, the current share of which is negligible. Grid operators had many misperceptions of VRE such as wind and solar energy, although its production cost has drastically dropped in recent years; solar photovoltaic farms' levelized cost of electricity (LCOE) dropped from US$0.378/kWh in 2010 to US$0.043/kWh in 2020 in some places [12]. Similarly, all LCOE cost trends for wind energy and concentrated solar power dropped drastically in 2010–2020 and will continue to drop in 2021 (Figure 3), but their share in the power generation mix remains small. Misperceptions stemmed from the concern that VRE production is variable and intermittent, and that its higher share in the grid will add costs as it will require backup capacity from conventional gas power plants [12].

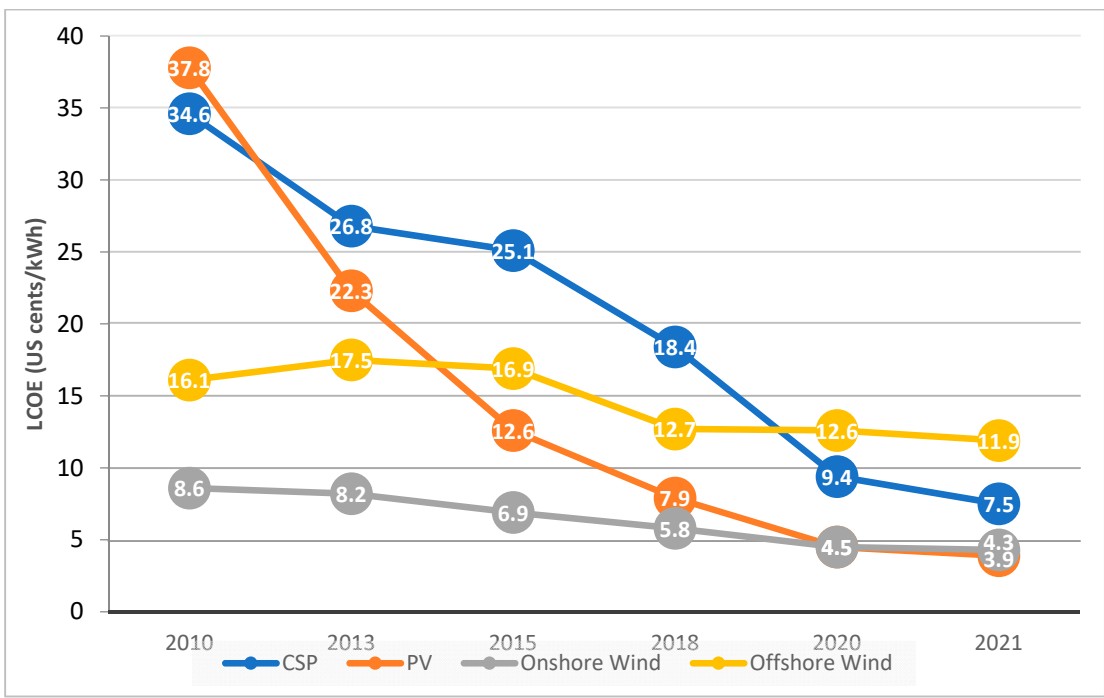

**Figure 3.** Falling Costs of Renewables in terms of Levelized Cost of Electricity (LCOE). CSP = concentrated solar power, kWh = kilowatt-hour, LCOE = levelized cost of electricity, PV = photovoltaic. Source: IRENA [13].

Technically, VRE power production output varies within a few seconds depending on wind or sunshine. However, the risk of variable energy output can be minimized if the power system is largely integrated within the country and within the region. The aggregation of output from solar and wind energy from different locations has a smoothing effect on net variability [12]. However, the ASEAN power grid is progressing slowly, and the integrated ASEAN power market might be far off because of several reasons, such as regulatory and technical harmonization issues within ASEAN power grids and utilities.

Scalable electricity production from wind and solar energy faces tremendous challenges from the current practice of system integration in ASEAN. Investors in solar or wind farms will confront high risks from electricity curtailment if surplus electricity is not used. Many countries have advanced research and technologies for battery storage (lithium-ion batteries) for surplus electricity produced from wind and solar energy, but advanced battery storage remains costly. Produced from electrolysis using surplus electricity, hydrogen has many advantages as it can be stored as liquid gas, which is suitable for numerous uses and easy to transport. Many ASEAN countries could produce wind, solar, hydropower, or geothermal electricity. Their resources, however, are far from demand centers and developing the resources would require large investments in undersea transmission cables. A solution would be to turn renewables into easily shipped hydrogen.

Hydrogen is a potential game changer for decarbonizing emissions, especially in sectors where they are hard to abate, such as cement and steel. Scalable resources from wind and solar energy and other renewables can be fully developed by widely adopting the hydrogen solution. The more electricity produced from wind and solar energy, the higher the penetration by renewables of the grid; at the same time, surplus electricity during low demand hours can be used to produce hydrogen. The more power generated from wind and solar energy and other renewables, the greater the possibility to increase the efficiency of electrolysis to produce hydrogen. On-site hydrogen production from wind and solar farms will solve the issue of curtailed wind and solar electricity. To increase the efficiency of electrolysis and allow further penetration by renewables of grids, a hybrid energy system including hydropower, geothermal, or nuclear plants, for example, would be the perfect energy choice. Since hydrogen is a

clean energy carrier and can be stored and transported for use in, amongst others, hydrogen vehicles, synthetic fuels, upgrading of oil and/or biomass, ammonia and/or fertilizer production, metal refining, heating, and other end uses, hydrogen development is an ideal pathway to a sustainable clean energy system and enables scalable VRE such as solar and wind energy.

### 4.3. Need to Reduce Renewable Hydrogen Production Cost

Cost-competitiveness of producing renewable hydrogen is key for the wide adoption of hydrogen uses. The upfront costs of renewable hydrogen such as electrolyzers, transport infrastructure, and storage, and the varying costs of electricity tariffs are key factors contributing to the high production cost of renewable hydrogen (Figure 4). 'Green' hydrogen production costs dropped drastically from US$10–US$15/kg of $H_2$ in 2010 to US$4–US$6/kg of $H_2$ in 2020, with varying assumptions of lower and higher upfront costs of electrolyzers with 20 MW and producing capacity of 4000 normal cubic meters per hour [3,13]. The costs are expected to reduce to US$2.00–US$2.60/kg of $H_2$ in 2030, which is competitive with steam methane reforming with CCS.

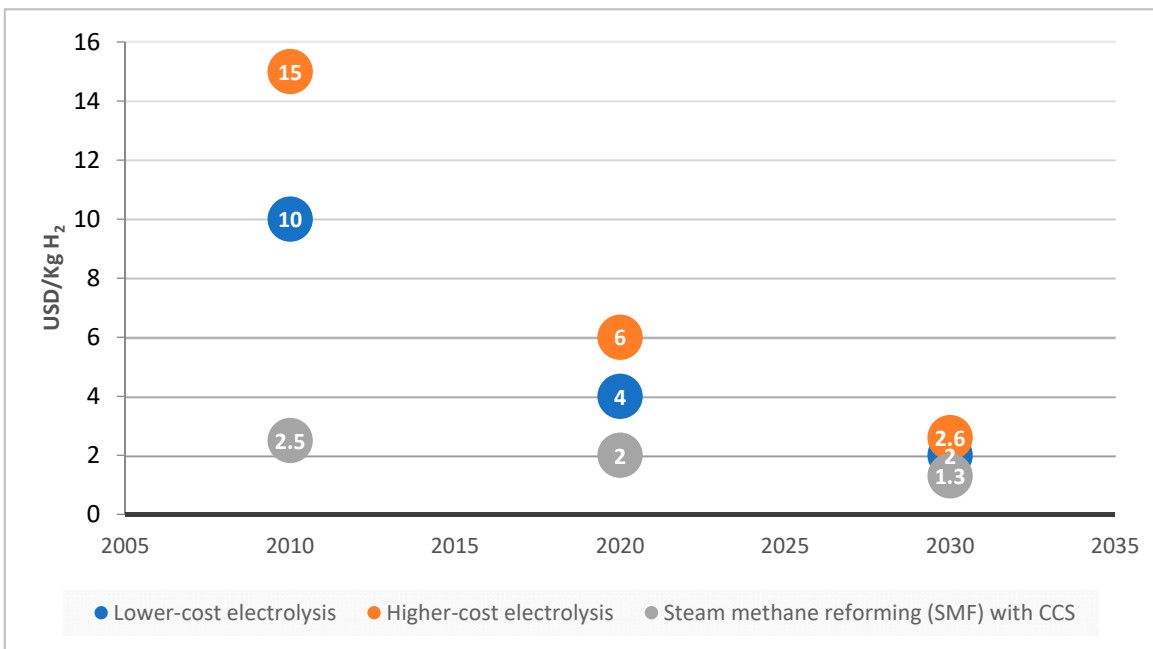

**Figure 4.** Hydrogen Production Cost Trends with Upfront Cost of Electrolyzers. CCS = carbon capture, sequestration, and storage, SMF = steam methane reforming. $H_2$ = hydrogen. Note: Assumption: 4000–normal cubic meter per hour (20 MW) polymer electrolyte membrane electrolyzers connected to offshore wind. The lower-cost electrolysis case is US$200/kilowatt (kW). The middle-cost electrolysis case is US$400/kW. The higher-cost electrolysis case is US$600/kW. Source: Authors, based on Hydrogen Council [4], DOE [14], and IRENA [15].

Considering the electricity tariffs of up to US$0.10/kWh with varying load factors of 10–50%, the cost of producing hydrogen ranged from US$0.90–US$5.50/kg of $H_2$ to US$4.20–US$8.90/kg of $H_2$ (Figure 5), meaning that electricity tariff is the major cost of producing hydrogen using electrolysis. At zero electricity tariff or when VRE is expected to be curtailed, the cost of producing hydrogen can be as low as US$0.90/kg of $H_2$ at an electrolyzer's load factor of 50%, and US$5.50/kg of $H_2$ at an electrolyzer's load factor of 10%. The International Renewable Energy Agency's target of cost-competitiveness of producing renewable hydrogen is US$2.00–US$2.50/kg of $H_2$ [16]. In this case, an electricity tariff of US$0.03/kWh with an electrolyzer's load factor of 30% is the most practical given all the constraints.

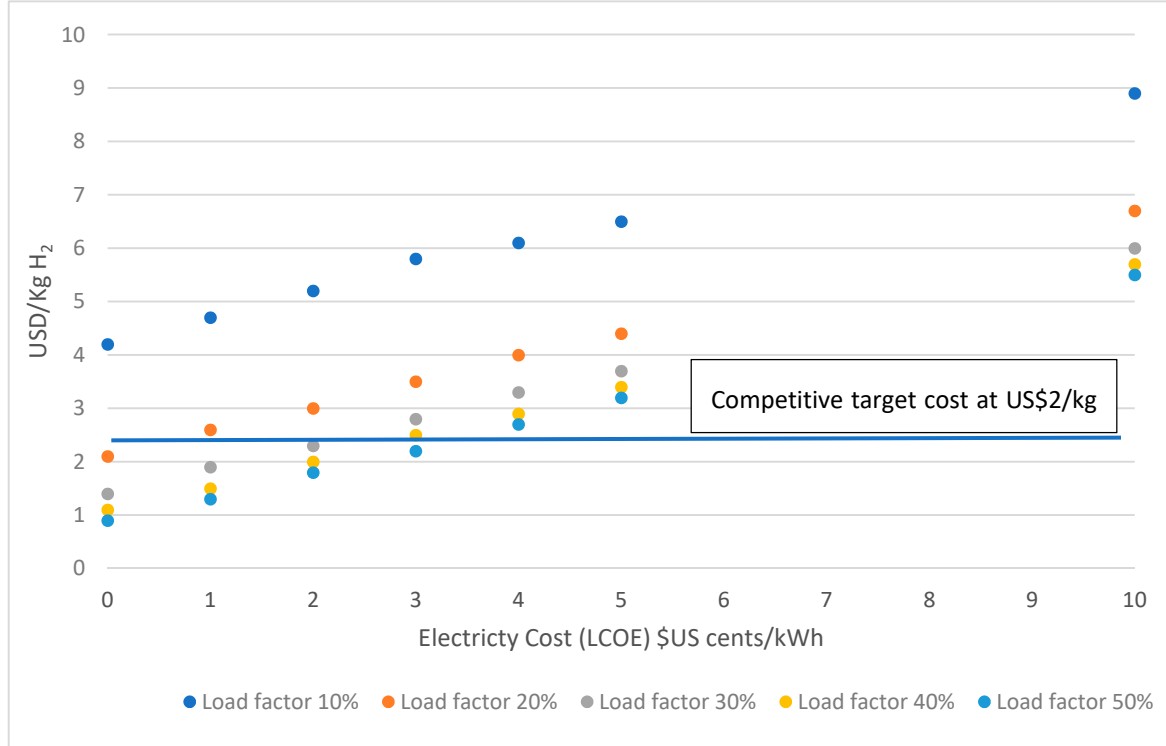

**Figure 5.** Hydrogen Production Cost with Varying Electricity Cost and Electrolysis Load Factors. $H_2$ = hydrogen, kWh = kilowatt-hour, LCOE = levelized cost of electricity. Note: Assumption: The polymer electrolyte membrane electrolyzer is connected with the grid. Source: Authors, based on Hydrogen Council [4], DOE [14], and IRENA [15].

The solar photovoltaic farm and onshore wind already cost US$0.02–US$0.03/kWh in some locations [16]. Even the target cost of US$2.00–2.50/kg of $H_2$ to produce 'green' hydrogen, however, would not be competitive with low-cost natural gas at US$5 per gigajoule (GJ) (Conversion factor: US$0.01/kWh = US$2.80/GJ.) (US$0.018/kWh), but would be with natural gas, which costs US$10–US$16/GJ (US$0.036–US$0.057/kWh).

Technically, if renewable hydrogen production uses only curtailed electricity from renewables, the operating load factor of electrolysis, which contributes the most to the cost of producing hydrogen, will likely be low at 10% or less. According to the Hydrogen Council [4], the electrolyzer will need to run at a load factor of at least 30% or more to lower the cost of producing hydrogen to US$2.00–2.50/kg of $H_2$, which is competitive with the natural gas grid price.

Electrolysis facilities must have a load factor above 30% to ensure the cost-competitiveness of producing renewable hydrogen, and other capital expenditures such as the electrolyzer's upfront cost must be reduced by 50% from US$840 today to US$420 per kilowatt by 2040. As wind and solar energy is expected to increase its share in the power generation mix, expected curtailed electricity from renewables will be higher by 10–30%. By 2030, the share of VRE curtailment will be 10–30% in Sweden, which provides the most incentives for renewable hydrogen [16]. In 2020, Chile, Australia, and Saudi Arabia have achieved the target cost of US$2.50/kg to produce 'green' hydrogen because of cheap access to electricity from wind and solar energy. The cost is expected to drop further to US$1.90/kg in 2025 and to US$1.20/kg in 2030, which is highly competitive with the cost of 'grey' hydrogen production.

Effective policies and incentives to develop and adopt hydrogen can promote economies of scale and cost-competitiveness in producing hydrogen, encouraging investors to manufacture electrolyzers; improve their efficiency, operation, and maintenance; and use low-cost renewable power such as hydrogen to enable scaling VRE penetration of the power grid. 'Green' hydrogen production cost could decline even faster and go even lower than US$2/kg of $H_2$ if governments, business, and stakeholders

join hands to adopt the wider use of 'green' hydrogen and increase investment and R&D in hydrogen fuels. Australia, Chile, and Saudi Arabia have achieved cost-competitiveness in wind and solar energy generation.

The energy transition will largely depend on the clean use of fossil fuel leading to a clean energy future. Although hydrogen is a clean fuel, the way it is produced matters. Almost 95% of hydrogen production is from natural gas with or without CCS. The gasification of coal can be used as feedstock for producing hydrogen, but it emits roughly four times more $CO_2$/kg of $H_2$ produced than natural gas feedstock does. The production cost of low-carbon 'blue' hydrogen depends on feedstock cost and suitable geographical CCS storage. IRENA (2019a) estimated that 'blue' hydrogen production in China and Austria with current CCS infrastructure could realize a production cost of about US$2.10/kg of $H_2$ for a cost of coal of about US$60 per ton. In the US, where natural gas is below US$3 per million British thermal units and has large-scale $CO_2$ storage such as depleted gas fields and suitable rock formations, 'blue' hydrogen cost could drop below US$1.50/kg in some locations. If the carbon cost of about US$50 per ton of $CO_2$ is considered, low-carbon hydrogen could reach parity with 'grey' hydrogen. 'Blue' hydrogen cost in the US and the Middle East could drop further to about US$1.20/kg in 2025 if economies of scale prevail.

World leaders need to provide a clear policy to develop and adopt hydrogen. The right policy will enable economies of scale for producing hydrogen cost-competitively, inducing investors to explore electrolyzer manufacturing; improve electrolyzer efficiency, operation, and maintenance; and use low-cost renewable power. With the full participation of governments, business, and stakeholders, hydrogen can become the fuel that enables scaling up renewable energy penetration in all sectors, decarbonizing global emissions.

*4.4. Need for Renewable Hydrogen Development Policies in ASEAN*

Until 2020, ASEAN did not have a hydrogen road map. APAEC, however, mentions alternative technologies and clean fuels such as hydrogen and energy storage. APAEC will help AMS increase the share of hydrogen in the energy mix. An ASEAN hydrogen road map is needed to guide national road maps. Based on the analysis of the drastic drop in the cost of VRE and electrolyzers, opportunities to introduce 'green' hydrogen produced using curtailed electricity will be plentiful. The hydrogen road map should include hydrogen development and penetration in transport, power generation, and industry. To guide investment, hydrogen penetration policies and targets must be set up. This study, however, can only suggest policies to develop, adopt, and use hydrogen. The study adopts Australia's hydrogen road map, especially its key polices [16,17], and tailors them to ASEAN's energy landscape. In developing the ASEAN's hydrogen roadmap, four key policies are highlighted below.

The first policy area is on financing which aims to provide access to lower-cost financing for hydrogen development and low-emission projects. In this regard, the government in ASEAN may need to consider providing fiscal policy incentives for local manufacturing for hydrogen development and financing incentives for low-emission electricity.

Another policy is on regulations which aims to set up targeted policies to stimulate hydrogen demand. In this regard, the government in ASEAN may need to consider developing hydrogen-specific regulations across AMS to support hydrogen development in power generation, transport, and industry. The regulation should allow grid-firming services from electrolyzers to be compensated, and allows for on-site hydrogen production and, where possible, position plants close to where the hydrogen will be used. Furthermore, the gas pipeline regulations should be reviewed to consider including gaseous hydrogen.

Thirdly, the policy is on research and development (R&D) which aims to establish demonstration projects for mature hydrogen technologies. The government in ASEAN should also consider setting up a hydrogen center of excellence as a research body to bring in all parties to work on technologies and policy coordination. The center should also conduct research and development in plant efficiency and safety, and in hydrogen shipment, pipeline, and storage.

The fourth policy target is on social acceptance which aims to develop a public engagement plan and strategy to support clean fuels such as hydrogen and ensures that communities understand all aspects of its use. The social acceptance is key for promoting willingness to pay for clean fuels.

To promote the future hydrogen society, ASEAN will have to develop a comprehensive hydrogen road map that includes a policy framework supporting hydrogen production, storage, and transport. The policy framework also needs to support hydrogen utilization in power generation, transport, heat production, industrial feedstock, and import and export. In developing the hydrogen road map, the governments in the countries of ASEAN should consult industrial, financial, and banking stakeholders. The road map will need to cultivate people's willingness to support a hydrogen society.

## 5. Conclusions and Policy Implications

ASEAN's energy transition will largely depend on increasing the share of renewables and clean fuels such as hydrogen and the clean use of fossil fuel to create a clean energy future. Fossil fuel (coal, oil, and gas) accounted for almost 80% of ASEAN's energy mix in 2017, a share that is expected to rise to 82% in BAU. Transitioning from a fossil fuel–based energy system to a clean energy system requires drastic policy changes to encourage embracing renewables and clean fuels whilst accelerating the use of clean technologies in employing fossil fuel (coal, oil, and natural gas). The study used energy modelling scenarios to explore policy options to abate emissions in ASEAN by giving wind and solar energy a high share of the energy mix and using electricity curtailment to promote renewable hydrogen production. The findings found that ASEAN has high potential to produce renewable hydrogen using curtailed electricity. The higher share of renewables under various policy scenarios will see a large reduction in $CO_2$ emissions, which could lead to decarbonizing emissions and contribute to abating global climate change. The potential emission abatement ranges from −340 Mt-C in APS to −648 Mt-C, −710 Mt-C, and −774 Mt-C in Scenario1, Scenario2, and Scenario3, respectively. Emissions will be cut by 28% from BAU to APS, 53% from BAU to Scenario1, 58% from BAU to Scenario2, and 64% from BAU to Scenario3.

The results of the study imply policy implications for ASEAN's energy policy reforms to ensure that clean fuels such as hydrogen and renewables and clean technologies will have a big role to play to decarbonize ASEAN's emissions. The below policy implications are derived from this study for the hydrogen adoption in ASEAN:

- ASEAN leaders must strongly commit to promoting a hydrogen society. ASEAN Ministers on Energy Meetings, facilitated by the ASEAN Secretariat, are an excellent platform for drafting a clear and actionable hydrogen development road map.
- ASEAN energy leaders must develop a clear strategy to promote hydrogen use in transport; power generation; and other sectors where emissions are hard to abate, such as the iron and steel industries. Singapore, Malaysia, Thailand, Indonesia, and the Philippines could take the lead by investing in R&D on hydrogen produced from renewables and non-renewables and by setting investment targets adapted from OECD countries. Investment in industries that can adopt hydrogen energy has strong potential, but to realize it ASEAN must accelerate its plans and strategies to embrace hydrogen use.
- Leaders in ASEAN and around the world must provide a clear investment policy to develop and adopt hydrogen as a fuel. The policy must enable economies of scale in cost-competitive production of hydrogen to induce investors to consider electrolyzer manufacturing; improvements in electrolyzer efficiency, operation, and maintenance; and the use of low-cost renewable power. With the full participation of governments, business, and stakeholders, hydrogen can become the fuel that enables scaling up renewable energy penetration in all sectors, decarbonizing global emissions.
- Governments must engage the public, build its awareness of the many benefits of a hydrogen society, and ensure that the public is willing to pay for them. The success of introducing hydrogen on a large scale needs the participation of all stakeholders, including governments and public

and private companies. Financing mechanisms such as banks must create favorable conditions to finance facilities such as electrolyzers. Governments must provide financial incentives to invest in developing hydrogen.

- Improving the electricity governance system in ASEAN developing countries will help reduce the cost of managing energy systems, allow the uptake of clean energy technology investment, and upgrade the grid system to bring in more renewables. The energy sector must be reformed; rules and procedures must allow more advanced and competitive technologies to enter the market. Electricity reform will attract foreign investment to modernize electricity infrastructure, including by making power systems more efficient and phasing out inefficient power generation and technologies.

- Unbundling of ownership in the electricity market, non-discriminatory third-party access to transmission and distribution networks, and the gradual removal of subsidies for fossil fuel–based power generation are key to ensure market competition. Other policies to attract foreign investment include tax holidays; reduction of market barriers and regulatory burdens; and plans to reduce the upfront cost investment, such as a rebate payment system through government subsidies and government guarantees that investment will be feasible and low risk.

**Author Contributions:** Conceptualization, F.K. and J.A.; methodology, H.P.; software, H.P.; validation, H.P., F.K. and J.A.; formal analysis, H.P.; investigation, H.P.; resources, H.P.; data curation, H.P.; writing—original draft preparation, H.P.; writing—review and editing, H.P.; visualization, H.P.; supervision, H.P.; project administration, H.P.; funding acquisition, H.P. All authors have read and agreed to the published version of the manuscript.

**Funding:** This research received no external funding.

**Conflicts of Interest:** The authors declare no conflict of interest.

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
