# Peer review of "Potential Renewable Hydrogen from Curtailed Electricity to Decarbonize ASEAN’s Emissions: Policy Implications"

_sustainability, doi:10.3390/su122410560_

Round 1
Reviewer 1 Report
- Overall the paper needs more concise writing. Too much general information provided, which anyone can easily find in a general news article. A research paper's focus should be to provide new analytical, simulated, or experimental results, rather than presenting too much general information.
- In the abstract section, line 12, 13 needs reference. However, adding a reference in the abstract does not look good. You can move those specific information to the introduction section and add references.
- In the abstract, write a few sentences on the key findings in this paper.
- Simplify the long sentence in-between lines 21-24 into small sentences.
- It’s difficult to follow the contents. The paragraphs should be coherent and subsequently build up the story that the authors want to tell. For example, in the Introduction section, there are lots of paragraphs presenting similar types of information. It’s very difficult to identify the focus of the paper. Similar information is again provided in section 3. Literature review.
I strongly suggest, combine, 1. Introduction, 2. Overall Objectives, 3. Literature Review as one single section as the introduction of the paper.
I suggest the authors construct the Introduction section with one or two paragraphs of background information, then a paragraph on the problem statement and current research gap, and finally a paragraph presenting the objective of this study.
- If the manuscript focuses on ASEAN countries, what is the point of discussing what’s going on in countries that are not ASEAN members? Please remove other countries' information and present information ASEAN members only.
- In section 3.1 there are lot’s of similar types of information. The section should be concise and added with the introduction section.
- In section 3.1 heaps of information given without any reference or source of those information. If you are presenting information from other sources, you need to provide reference for each data.
- The information provided in Table 4 do not qualify as a table, they should be written in a paragraph. Section 5 is the core section of this manuscript. This section should only present what you have identified from your analysis, instead of other secondary source data, as presented in section 5.2. Instead of table 4, the suggestions should be discussed in this section in one or two paragraphs. I suggest re-write section 5, concise the contents, present your own finding and suggestions rather than delivering data from other sources.
- Figure 3, 4 no vertical axis scale, please add.
- In line 119, what is ‘variable renewable energy (VRE)’? Explain it.
- Center align all equations and add equation number.
- Similarly, authors need to concise the ‘conclusion’ section. Too many unnecessary information provided in the ‘conclusion’. A reader will be lost to identify the findings from this paper. The conclusion should include just few paragraphs presenting the critical findings of this research study.
Author Response
Dear Prof/Dr. (Reviewers),
We are very grateful for your time providing valuable comments for our paper. We really appreciate this. We made our best efforts to address each of your comments and suggestions. The paper has been revised substantially. First, we have revised the abstract and added the key research findings and policy implications. For the article, we have combined the sections of Introduction, objectives, and literature reviews to be under “Introduction section”. We have also recognized the Introduction section carefully to ensure the flow/story line that we want to tell about this topic of research. We also removed the part that is not relevant to ASEAN context as suggested in the comments.
For the table 4 of the section about ASEAN’s hydrogen road map, we have removed the table 4 and wrote it as a policy direction necessarily for the establishment of ASEAN’s hydrogen roadmap. You will find it very interesting in this revised paper. We also addressed the remaining minor comments and suggestions to ensure the readers will understand this paper well.
Furthermore, we have also addressed comments and suggestions from other reviewers in this process.
We hope that you will kindly consider this paper as we believed that the paper will have impacted positively for ASEAN’s stakeholders to prepare for the hydrogen society.
Again, thank you so much for your valuable time spent generously commenting on our paper.
Best regards,
Han Phoumin
Fukunari Kimura
Jun Arima
Reviewer 2 Report
The article addresses an important and actual topic of decarbonising emissions, exploring the scalability of hydrogen production from renewable sources. Undoubtedly, this is an important step towards "cleaner energy".
However, I have many comments to the manuscript of an editorial and substantive nature.
Editorial
In the manuscript I noticed many minor errors (units used, wrong font, too large spaces between words, wrong numbering of figures, etc.):
Line 78: kilowatt – it should be kW
Line 81: kilogram – it should be kg
Line 106: different font than it should be
Line 108, 110: too much space between words
Line 202: degrees Celsius = oC
Line 195: H2 – it should be H2
The font on formulas in section 3.2 is not correct; the numbering of formulas (1) etc. is also missing
Table 2. H2 – it should be H2
Line 365: CO2 – it should be CO2
Figure 3: please add a vertical axis along with its name
Line 432: megawatts = MW
Line 438: megawatt = MW
Figure 4 occurs twice; in the second Figure 4, values for the dots are missing
Line 488, 493, 495, 551: CO2 – it should be CO2
Table 4 – table not formatted correctly; Tables 1-3 are also not formatted correctly; RD&D - it should be R&D
Lines 561-592 – not correct font
The manuscript also uses the wrong format of the footnotes - it should have the following form: [1], [2] etc. However, the Harvard style was used.
I recommend that you check the text very carefully in terms of editing and language.
Substantive
The abstract is a bit too long (over 200 words) and shows too much background. I think it would be good to rewrite the abstract again taking into account the following recommendations:
A single paragraph of about 200 words maximum. For research articles, abstracts should give a pertinent overview of the work. We strongly encourage authors to use the following style of structured abstracts, but without headings: 1) Background: Place the question addressed in a broad context and highlight the purpose of the study; 2) Methods: Describe briefly the main methods or treatments applied; 3) Results: Summarize the article's main findings; and 4) Conclusions: Indicate the main conclusions or interpretations. The abstract should be an objective representation of the article, it must not contain results which are not presented and substantiated in the main text and should not exaggerate the main conclusions.
Points 2, 3 and 4 should be given more emphasis.
JEL codes are not needed, and from keywords I would remove 'system integration'.
The introduction presents an outline of the discussed research problem, its justification and significance. However, there is no reference to the literature here, there are very few footnotes. There are also no research questions, thesis / hypothesis or article objectives.
In chapter 2, it is not known on what basis the study's objectives and structure were formulated. It would be good to combine chapters 1 and 2 so that they form a coherent whole.
The Literature Review section contains very few references and does not present a scientific view of the topic in the light of the actual literature. It is not known what has been done on this topic so far.
Chapter 3.2 I don't know if it's needed at all, Sustainability it's not a chemical journal.
Methodology and Scenario Assumptions could be extended further, especially the rationale for choosing Long-range Energy Alternative Planning System (LEAP) software. What are the other software? How did the adopted scenario assumptions affect the model result? Why were these 3 scenarios chosen?
The results and discussion were presented correctly, although it is not clear how the individual values were obtained. It would be good to describe how the software generated the individual results.
There was no comparative analysis of 3 scenarios. There are also no directions for possible further research and no information about the limitations of the simulations. It was also possible to provide the application possibilities of the proposed approach. I also failed to find how the solution proposed by the authors stands out from the others. What are its advantages?
The references are up-to-date, but a large part is online. Maybe it's worth supplementing them with books and articles?
The text certainly has some advantages, but I think that in order to publish it, it would be good to introduce all the suggested corrections.
Author Response
Dear Prof/Dr. (Reviewers),
We are very grateful for your time providing valuable comments for our paper. We really appreciate this. We made our best efforts to address each of your comments and suggestions.
We have carefully addressed your comments in terms of minor errors. We have try our best to address your comments as well as comments made by other reviewers.
The paper has been revised substantially. First, we have revised the abstract and added the key research findings and policy implications. For the article, we have combined the sections of Introduction, objectives, and literature reviews to be under “Introduction section”. We have also recognized the Introduction section carefully to ensure the flow/story line that we want to tell about this topic of research. We also removed the part that is not relevant to ASEAN context as suggested in the comments.
For the table 4 of the section about ASEAN’s hydrogen road map, we have removed the table 4 and wrote it as a policy direction necessarily for the establishment of ASEAN’s hydrogen roadmap. You will find it very interesting in this revised paper. We also addressed the remaining minor comments and suggestions to ensure the readers will understand this paper well.
We hope that you will kindly consider this paper as we believed that the paper will have impacted positively for ASEAN’s stakeholders to prepare for the hydrogen society.
Again, thank you so much for your valuable time spent generously commenting on our paper.
Best regards,
Han Phoumin
Fukunari Kimura
Jun Arima
Round 2
Reviewer 1 Report
The revised manuscript showing all the corrections and editing. That is fine. Besides this, authors also should have added a neat and clean copy of the manuscript. It could be easier to review the manuscript that way.
No point to point reply from authors to my review is added in this submission. Authors should have added a point by point reply to my review.
All figures letters and numbers are in grey colour, please check it. All text and numbers should be black colour.
In section 2, chemical reactions are referred as process (1) , process (2)...etc.
The word 'process' is not necessary in there, just add the equation number.
Use MDPI equation fonts for all equations.
Author Response
Dear Prof/Dr. (Reviewers),
Again, we are very grateful for your further comments on our paper. This time, we have done the track change, so that you can find it easy to check our responses to your comments. Since, the 2nd round of comments are minor, we are pleased to let you know that we have improved the paper as follows:
- Add more citations and follow the journal style of reference. We spent a lot of time on this part as to ensure the citations and references are in-lined with the journal requirement.
- For the equations, we have changed it to the tyle of the journal
- We add additional para to explain the reason why this research employs the LEAP to estimate the future power generation demand in ASEAN.
Thank you very much.
Best regards,
Han Phoumin
Fukunari Kimura
Jun Arima
Reviewer 2 Report
The authors revised the manuscript in line with my comments. Below is a reference to the introduced changes:
Editorial
In the manuscript I noticed many minor errors (units used, wrong font, too large spaces between words, wrong numbering of figures, etc.):
Line 78: kilowatt – it should be kW - corrected
Line 81: kilogram – it should be kg - corrected
Line 106: different font than it should be - corrected
Line 108, 110: too much space between words - corrected
Line 202: degrees Celsius = oC - corrected
Line 195: H2 – it should be H2 - corrected
The font on formulas in section 3.2 is not correct; the numbering of formulas (1) etc. is also missing - corrected
Table 2. H2 – it should be H2 - corrected
Line 365: CO2 – it should be CO2 - corrected
Figure 3: please add a vertical axis along with its name - corrected
Line 432: megawatts = MW - corrected
Line 438: megawatt = MW - corrected
Figure 4 occurs twice; in the second Figure 4, values for the dots are missing - corrected
Line 488, 493, 495, 551: CO2 – it should be CO2 - corrected
Table 4 – table not formatted correctly; Tables 1-3 are also not formatted correctly; RD&D - it should be R&D
Lines 561-592 – not correct font - corrected
The manuscript also uses the wrong format of the footnotes - it should have the following form: [1], [2] etc. However, the Harvard style was used.
The manuscript has been corrected in terms of editing and language. However, I would recommend checking Sustainability journal guidelines for e.g. figure captions, tables or footnotes.
I assess the introduced editorial changes positively.
Substantive
The abstract is a bit too long (over 200 words) and shows too much background. I think it would be good to rewrite the abstract again taking into account the following recommendations:
A single paragraph of about 200 words maximum. For research articles, abstracts should give a pertinent overview of the work. We strongly encourage authors to use the following style of structured abstracts, but without headings: 1) Background: Place the question addressed in a broad context and highlight the purpose of the study; 2) Methods: Describe briefly the main methods or treatments applied; 3) Results: Summarize the article's main findings; and 4) Conclusions: Indicate the main conclusions or interpretations. The abstract should be an objective representation of the article, it must not contain results which are not presented and substantiated in the main text and should not exaggerate the main conclusions.
Points 2, 3 and 4 should be given more emphasis.
JEL codes are not needed, and from keywords I would remove 'system integration'.
The abstract was corrected and shortened (less than 200 words). Main results and conclusions were also added.
The introduction presents an outline of the discussed research problem, its justification and significance. However, there is no reference to the literature here, there are very few footnotes. There are also no research questions, thesis / hypothesis or article objectives.
In chapter 2, it is not known on what basis the study's objectives and structure were formulated. It would be good to combine chapters 1 and 2 so that they form a coherent whole.
The Literature Review section contains very few references and does not present a scientific view of the topic in the light of the actual literature. It is not known what has been done on this topic so far.
Chapter 3.2 I don't know if it's needed at all, Sustainability it's not a chemical journal.
Chapters 1, 2 and 3.1 were organized and merged. The text is now clearer and easier to read. Analyzes were limited to ASEAN countries. One reference added: (Han, Kimura, and Arima, 2020).
Methodology and Scenario Assumptions could be extended further, especially the rationale for choosing Long-range Energy Alternative Planning System (LEAP) software. What are the other software? How did the adopted scenario assumptions affect the model result? Why were these 3 scenarios chosen?
No changes have been made to this section.
The results and discussion were presented correctly, although it is not clear how the individual values were obtained. It would be good to describe how the software generated the individual results.
No changes have been made to this section.
There was no comparative analysis of 3 scenarios. There are also no directions for possible further research and no information about the limitations of the simulations. It was also possible to provide the application possibilities of the proposed approach. I also failed to find how the solution proposed by the authors stands out from the others. What are its advantages?
Table 4 has been replaced with text to better understand the four key policies for the development of the ASEAN's hydrogen roadmap. However, the additions suggested by the reviewer were not presented, regarding, among others: comparative analysis of 3 scenarios, directions for possible further research, research limitations, advantages and application possibilities of the proposed approach.
The references are up-to-date, but a large part is online. Maybe it's worth supplementing them with books and articles?
One reference has been added: Han, P., Kimura, F., and Arima. J (2020), which proves that the authors conduct research in this research area.
The authors took into account only some of the reviewer's comments. Manuscript now looks better and can be published after supplementing with the indicated elements (red color).
Author Response
Dear Prof/Dr. (Reviewers),
Again, we are very grateful for your further comments on our paper. This time, we have done the track change, so that you can find it easy to check our responses to your comments. Since, the 2nd round of comments are minor, we are pleased to let you know that we have improved the paper as follows:
- Add more citations and follow the journal style of reference. We spent a lot of time on this part as to ensure the citations and references are in-lined with the journal requirement.
- For the equations, we have changed it to the style of the journal
- We add additional para to explain the reason why this research employs the LEAP to estimate the future power generation demand in ASEAN.
Thank you very much.
Best regards,
Han Phoumin
Fukunari Kimura
Jun Arima